# Stratification of Gut Microbiota Profiling Based on Autism Neuropsychological Assessments

**DOI:** 10.3390/microorganisms12102041

**Published:** 2024-10-09

**Authors:** Chiara Marangelo, Pamela Vernocchi, Federica Del Chierico, Matteo Scanu, Riccardo Marsiglia, Emanuela Petrolo, Elisa Fucà, Silvia Guerrera, Giovanni Valeri, Stefano Vicari, Lorenza Putignani

**Affiliations:** 1Research Unit of Microbiome, Bambino Gesù Children’s Hospital, IRCCS, 00165 Rome, Italy; chiara.marangelo@opbg.net (C.M.); pamela.vernocchi@opbg.net (P.V.); federica.delchierico@opbg.net (F.D.C.); matteo.scanu@opbg.net (M.S.); riccardo.marsiglia@opbg.net (R.M.); 2Child and Adolescent Neuropsychiatry Unit, Bambino Gesù Children’s Hospital, IRCCS, 00165 Rome, Italy; emanuela.petrolo@opbg.net (E.P.); elisa.fuca@opbg.net (E.F.); silvia.guerrera@opbg.net (S.G.); giovanni.valeri@opbg.net (G.V.); or stefano.vicari@unicatt.it (S.V.); 3Life Sciences and Public Health Department, Università Cattolica del Sacro Cuore, 00168 Rome, Italy; 4Unit of Microbiomics and Research Unit of Human Microbiome, Bambino Gesù Children’s Hospital, IRCCS, 00146 Rome, Italy

**Keywords:** autism, neuropsychological phenotypes, autism severity, gut microbiota, microbial dysbiosis index (MDI)

## Abstract

Autism spectrum disorder (ASD) is a neurodevelopmental disorder. Investigations of gut microbiota (GM) play an important role in deciphering disease severity and symptoms. Overall, we stratified 70 ASD patients by neuropsychological assessment, based on Calibrated Severity Scores (CSSs) of the Autism Diagnostic Observation Schedule-Second edition (ADOS-2), Child Behavior Checklist (CBCL) and intelligent quotient/developmental quotient (IQ/DQ) parameters. Hence, metataxonomy and PICRUSt-based KEGG predictions of fecal GM were assessed for each clinical subset. Here, 60% of ASD patients showed mild to moderate autism, while the remaining 40% showed severe symptoms; 23% showed no clinical symptoms, 21% had a risk of behavior problems and 56% had clinical symptoms based on the CBCL, which assesses internalizing problems; further, 52% had no clinical symptoms, 21% showed risk, and 26% had clinical symptoms classified by CBCL externalizing problems. Considering the total CBCL index, 34% showed no clinical symptoms, 13% showed risk, and 52% had clinical symptoms. Here, 70% of ASD patients showed cognitive impairment/developmental delay (CI/DD). The GM of ASDs with severe autism was characterized by an increase in *Veillonella*, a decrease in *Monoglobus pectinilyticus* and a higher microbial dysbiosis index (MDI) when compared to mild-moderate ASDs. Patients at risk for behavior problems and showing clinical symptoms were characterized by a GM with an increase of *Clostridium*, *Eggerthella*, *Blautia*, *Intestinibacter*, *Coprococcus*, *Ruminococcus*, *Onthenecus* and *Bariatricus*, respectively. Peptidoglycan biosynthesis and biofilm formation KEGGs characterized patients with clinical symptoms, while potential microbiota-activated PPAR-γ-signaling was seen in CI/DD patients. This evidence derived from GM profiling may be used to further improve ASD understanding, leasing to a better comprehension of the neurological phenotype.

## 1. Introduction

Autism spectrum disorder (ASD) is a neurodevelopmental disorder characterized by several core symptoms such as social communications and interaction deficits, repetitive patterns of behavior, restricted interests and altered sensory processing [1]. This is a multi-factorial disorder [2], and its development could be associated with genetic factors, immune modulation diseases, inflammation, and exposure to environmental toxins [3]. Individuals with ASD frequently exhibit a range of co-occurring conditions such as gastrointestinal (GI) symptoms, including constipation, diarrhea, and abdominal pain [4], and psychopathological symptoms such as irritability, anxiety and affective disorders, dysregulation and externalizing [5]. Although the etiopathogenesis of ASD is not fully understood, recent evidence has highlighted an alteration in the gut microbiota (GM) composition in ASD patients compared with neurotypical individuals [6]. Physiologically, the intestinal microorganisms communicate with the central nervous systems (CNS) through a complex bidirectional pathway (gut–brain axis), which plays a key role in brain development and homeostasis [7]. Indeed, the community of microorganisms in the GI tract has the ability to influence brain physiology and social behavior through multiple pathways, such as immune activation, as well as the production of microbial metabolites, peptides, and different neuromodulators and neurotransmitters [8].

The transition from a symbiotic to a dysbiotic condition causes an intestinal imbalance that may affect other organs and tissues such as liver, lung, adipose tissue, and direct immune balance [9]. Generally, intestinal dysbiosis may lead to a decrease in bacterial diversity and to a production of pro-inflammatory molecules and microbial metabolites, causing functional, structural, and metabolic impairments within the gut ecosystem [10]. Increasing evidence indicates GM imbalance as a possible concurring factor related to ASD severity and other neurobehavioral disorders [11,12].

However, the association amongst GM, ASD severity and neurological symptoms remains largely unexplored.

The aim of the present study was to characterize GM profiles in pediatric ASD patients, focusing on ASD neuropsychological assessment stratification, driven by the Calibrated Severity Scores (CSS) of the Autism Diagnostic Observation Schedule-Second edition (ADOS-2), the Child Behavior Checklist (CBCL) and intelligent quotient/developmental quotient (IQ/DQ) indexes. Finally, for each of the ASD neuropsychological-dependent subgroups, a Microbial Dysbiosis Index (MDI) was used to provide a quantitative evaluation of GM dysbiosis.

## 2. Materials and Methods

### 2.1. Patient Characteristics and Sample Collection

A cohort of 70 patients with the diagnosis of ASD was recruited at the Bambino Gesù Children’s Hospital (Rome, Italy). This study group, aged 3–17 years (average age, 6.91 years, SD ± 3.17), included 56 males and 14 females.

The inclusion criteria were as follows: age between 3 and 17 years; the diagnosis of ASD supported by the use of gold standard instruments (see Section 2.2.)*;* the absence of detected co-occurring neurological conditions (e.g., epilepsy) and genetic syndromes; patients who did not complete the neuropsychological assessment because of limited compliance.

The diagnosis of ASD was established according to criteria of the Diagnostic and Statistical Manual of Mental Disorders, fifth edition (DSM-5) [13], and confirmed by the ADOS-2 and, when available, the Autism Diagnostic Interview—Revised (ADI-R).

### 2.2. Clinical and Neuropsychological Assessment

ASD severity and symptoms were evaluated using the ADOS-2 [14], considered the “gold standard” in the assessment of communication, social interaction and restricted and repetitive behaviors for individuals with a diagnosis of ASD [14,15,16].

The ADOS-2 is an activity-based assessment administered by trained clinicians to observe the presence of ASD symptoms in a subject’s behavior; it allows the assessment of the severity levels of the core ASD symptoms. In particular, to measure the severity of ASD symptomatology, the instrument includes a 10-point severity metric known as CSS [17,18,19], which limits the impacts of other characteristics when assessing symptom presentation in individuals of the same age and language abilities, allowing a non-biased assessment of the autism core symptoms [20].

Based on the CSS, the patients were divided into two levels of severity according to the ADOS scores criteria (i.e., mild/moderate and severe autism) [1].

The ADI-R [21] is a parent-report semi-structured interview used for establishing a clinical diagnosis of ASD. It follows the DSM-IV-TR diagnostic criteria for ASD in children with a mental age of 18 months and above [22]. The ADI-R produces algorithm scores for each of the three subdomains related to autistic symptoms: qualitative deficits in reciprocal social interactions, qualitative communication abnormalities, and a limited range of interests and/or stereotypic behaviors.

Behavioral and psychopathological screening was performed through the CBCL [23,24], a questionnaire completed by parents/caregivers reporting their evaluation of their child’s behavior during the preceding 6 months on a 3-point Likert scale for each item (0 = not true; 1 = somewhat or sometimes true; 2 = very true or often true). The CBCL for ages 1.5 to 5 consists of 100 problem items. There are seven syndrome scales: Emotionally Reactive, Anxious/Depressed, Somatic Complaints, Withdrawn, Sleep Problems, Attention Problems, and Aggressive Behavior. The summary profile contains the Internalizing, Externalizing, and Total Problems scales. Finally, there are five DSM-oriented scale profiles, consistent with the diagnostic categories of DSM-IV-TR (Affective Problems, Anxiety Problems, Pervasive Developmental Problems, Attention Deficit/Hyperactivity Problems, and Oppositional Defiant Problems) [22]. The CBCL 6–18 is a 113-item parent-report questionnaire that generates eight syndrome scales (Anxious/depressed, Withdrawn/depressed, Somatic complaints, Social problems, Thought problems, Attention problems, Rule-breaking behaviors, and Aggressive behaviors), and two broad-band scales (Internalizing problems and Externalizing problems). The sum of all the items generates the “Total Problem” scale. Six DSM-oriented scales are also included (Affective, Anxiety, Somatic, Attention-deficit/hyperactivity disorder (ADHD), Oppositional defiant problems, and Conduct problems).

For the current study, we considered the two broad-band scales and the “Total Problems” scale.

According to the range of the CBCL, patients were stratified into three categories as: t-score < 60 (non-clinical symptoms); t-score between 60 and 63 (child at risk for problem behaviors); and t-score ≥ 64 (clinical symptoms).

The cognitive level (intelligent quotient/developmental quotient; IQ/DQ) was assessed according to age, language level, and attentional resources through the Wechsler Intelligence Scale for Children–Fourth Edition (WISC-IV) [25]. WISC-IV administration provides four different indexes: Verbal Comprehension Index; Perceptual Reasoning Index; Working Memory Index; and Processing Speed Index. In cases of language problems, we administered the Leiter International Performance Scale—3rd Edition (Leiter-3) [26], which offers a nonverbal assessment of intelligence, evaluating reasoning abilities through analogy, matching, and general perceptual reasoning, without being influenced by language skills or formal education. The Global Non-Verbal Intelligent Quotient obtained through this test is based on four subtests: Figure Ground; Form Completion; Classification and Analogies; and Sequential Order. The Griffiths III [27] was administered to assess developmental quotient. The instrument provides an overall measure of a child’s development, as well as an individual profile of strengths and needs across five areas: Foundations of Learning; Language and Communication; Eye and Hand Coordination; Personal–Social–Emotional; Gross Motor.

The metadata of ASD patients, recorded at the time of enrolment, included anamnestic data and clinical parameters such as neuropsychological features reported as autism symptoms severity, cognitive level and behavioral problems (Appendix A). Based on these clinical variables, patients were stratified into subgroups.

Fecal samples were stored at −80 °C at the OPBG Microbiome Unit until GM analyses.

### 2.3. Ethics Statement

The study was approved by the OPBG Ethics Committee (1404_OPBG_2017) and conducted in accordance with the Principles of Good Clinical Practice and the Declaration of Helsinki. Written informed consent was obtained from either the parents or the legal representative of the children.

### 2.4. GM Metataxonomy Profile: Wet and Dry Analyses

Bacterial DNA was extracted from 200 mg of stools, using the QIAmp Fast DNA Stool mini kit (Qiagen, Hilden, Germany), following the manufacturer’s instructions. The V3–V4 variable regions (~ 460 bp) of the 16S rRNA locus were amplified following the MiSeq rRNA Amplicon Sequencing protocol (Illumina, San Diego, CA, USA). The DNA libraries were obtained using a 2 × KAPA Hifi HotStart ready Mix (KAPA Biosystems Inc., Wilmington, MA, USA), following the manufacturer’s protocol. Illumina Nextera adaptor-primers were used to barcode each sample. The final bacterial libraries were quantified by Quant-iT™ PicoGreen^®^ dsDNA Assay Kit (Thermo Fisher Scientific, Waltham, MA, USA). Samples were run by a next generation sequencing (NGS) approach on an Illumina MiSeq™ platform, according to the manufacturer’s specifications, to generate paired-end reads of 300 base-length.

A total of 140 fastq files were imported into QIIME2 v2023.2 for the 70 samples [28]. The Quality Check (QC) was performed by including only samples that had a depth greater than 1000 reads and a minimum Phred score exceeding 20. By means of DADA2 [29], paired-end reads were filtered from chimeras and assembled into Amplicon Sequence Variants (ASVs) with a cut off of 99% similarity. ASVs were assigned taxonomically by Greengenes2 algorithm through Greengenes nucleotide sequence database v2022.10. Sequences labeled with Mitochondria, Chloroplast and Eukaryota were filtered. The ASV table was filtered out, retaining the ASV present in at least the 25% of the total samples. Moreover, a filter based on the ASV relative abundance, retaining ASV with a relative abundance > 0.01, was applied. The phylogenetic tree was built with the phylogeny align-to-tree-mafft-fasttree method, which is based on a de novo approach [30].

### 2.5. Microbial Dysbiosis Index (MDI) Calculation

Gut dysbiosis was calculated by an OPBG algorithm (patent no. WO2017216820A1, https://patents.google.com/patent/WO2017216820A1/en, accessed on 21 December 2017) based on a metagenomic method for the in vitro diagnosis of microbiota dysbiosis. A degree of dysbiosis in ASD patients was assigned by comparing the ASD microbiota profiles with those from healthy subjects (controls, CTRLs) matched for age and gender. Based on the percentage quadratic dissimilarity index, Z = (½ × Σ(f_case_ − f_controls_)2)½ × 100 [31], calculated using the median value of the GM taxa distribution of the patient (f_case_) and the median value of GM taxa distribution of CTRLs (f_controls_), at the phylum, family and genus taxonomic levels. The MDI ranged between 0 (indicates no dissimilarity) and 1 (indicates maximum dissimilarity), and could be expressed as a percentage [32].

### 2.6. Statistical Analysis

All statistical analyses were performed with R software v4.3.0.

Feature table of absolute frequency, rarefied to the low-number of sequences in a sample, was used for α- and β-diversity analyses (R package, “Phyloseq”). Statistical analyses based on Kruskal–Wallis and Mann–Whitney tests were applied to compare the results of α-diversity with Chao1, Shannon and Simpson indices.

Beta diversity was assessed using the Bray–Curtis algorithm to capture the dissimilarity between microbiome profiles, compared with the Permutational multivariate analysis of variance (PERMANOVA). Principal coordinate analyses (PCoA) plots was used to visualize the beta diversity of samples clustered in groups.

For further analyses, ASVs abundances from each sample were normalized with the Cumulative Sum Scaling method [33] and the ASV table was filtered, retaining the ASVs present in at least the 25% of the total samples and with relative abundance > 0.01.

The relative abundance of taxa was represented with a bar plot (R package, “ggplot2”) for different taxonomic levels.

The data normality distributions were evaluated using the Shapiro–Wilk test, and non-parametric Mann–Whitney and Kruskal–Wallis tests were applied to evaluate the difference in the microbiota at the phylum (L2), family (L5), genus (L6) and species (L7) levels between sample groups. In particular, two analyses were performed on ASV tables at the genus and species levels to identify “key” biomarkers that discriminate ASD groups stratified by different clinical features: Linear Discriminant Analysis Effect size (LEfSe) [34] and linear regression models corrected by confounding factors. ASVs yielded by both methods were statistically significantly different between the ASD groups (Linear discriminant analysis (LDA) > 2 and *p*-value < 0.05 for LEfSe and *p*-value < 0.05 for linear model) were considered key discriminatory biomarkers specific for the clinical variables considered.

Potential confounding factors were evaluated through anamnestic features, such as gender and age, which were included in the linear model to filter bacteria that were dependent on them.

Based on the KEGG Orthology database (https://www.genome.jp/kegg) [35], the Phylogenetic Investigation of Communities by Reconstruction of Unobserved States of Correlation 2 (PICRUSt2) software v2.4.1 was used to predict functional pathways [36]. To identify statistically significantly (*p*-value ≤ 0.05) different KO pathways, the LEfSe method and linear models (limma voom method in R package “ggpicrust2”) were applied [37].

## 3. Results

### 3.1. Participants Characteristics and Neuropsychological Features

In this study, 70 ASD patients were enrolled (Figure 1). The ASD cohort included 80% (56/70) males and 20% (14/70) females (Figure 1) with a mean age of 6.91 ± 3.17. Considering the CSS (for 5 patients, the information was missing), with a mean value of 6.98 ± 1.47, 60% of ASD patients (39/65) belonged to the mild-to-moderate autism range, while the remaining 40% (26/65) fit into the severe autism range (Figure 1).

The scores obtained from the Internalizing (CBCL_INT), Externalizing (CBCL_EXT), and Total (CBCL_TOT) CBCL Scales evidence that, in general, a large part of the participants presented clinical symptoms. Regarding CBCL indices (for 9 patients the information was missing), particularly CBCL_INT, 23% of patients (14/61) reported no clinical symptoms, 21% (13/61) showed a risk of behavior problems and 56% (34/61) showed clinical symptoms; based on the CBCL_EXT, 52% (32/61) of patients reported no clinical symptoms, 21% (13/61) showed a risk of behavior problems, and 26% (16/61) showed clinical symptoms; lastly, based on CBCL_TOT, 34% of patients (21/61) showed no clinical symptoms, 13% (8/61) showed a risk of behavior problems, and 53% (32/61) presented clinical symptoms (Figure 1).

Stratifying by intelligence measures (for 1 patient the information was missing), reported by IQ/DQ (mean value 62.9 ± 19.9), 70% (48/69) of ASD patients showed cognitive impairment/developmental delay (CI/DD), while 30% (21/69) did not (Figure 1).

### 3.2. Metataxonomic GM Profile

NGS sequencing analysis generated 7,180,111 sequences obtained from all 70 samples, representing 1802 ASVs.

For each patient, the MDI was calculated to assay the level of gut dysbiosis. The MDI mean value for the ASD cohort was 23.8 ± 8.76. In the patient GM profile, the main phyla were Firmicutes_A (45%), Bacteroidota (25.4%), Verrucomicrobiota (10%), Actinobacteria (7.6%) and Proteobacteria (5%). The most abundant families were as follows: Bacteroidaceae (20.7%), Lachnospiraceae (15.7%), Ruminococcaeae (15%) and Akkermansiaceae (10.6%), and the main genera were *Bacteroides_H* (14%), *Faecalibacterium* (10.2%), *Phocaeicola_A_858004* (9.1%) and *Akkermansia* (6.4%).

The ASD cohort was categorized by age and gender to assess their potential influence as confounding factors on the GM profile. Regarding age, α-diversity showed no statistically significant results (Appendix A) while β-diversity analysis showed a statistically significant clustering (PERMANOVA *p*-value < 0.05) of subjects stratified for age class, confirming the variable “age” as a confounding factor (Figure 2). Conversely, considering the gender variable, both ecological analyses revealed no statistically significant results, indicating that gender was not a confounding factor (Appendix A).

### 3.3. Characterization of Participants’ GM Profiles in Relation to Neuropsychological Features

The ecological analyses, with results reported as α- and β-diversity, were performed to evaluate the GM bacterial compositions of the ASD patients in relation to the clinical data.

Based on CBCL_EXT, α-diversity, computed by Shannon, Simpson and Chao1 indices, showed a significant (*p*-value < 0.05) increase in diversity in patients with clinical symptoms compared to patients without clinical symptoms; higher values of α-diversity (*p*-value < 0.05), based on Shannon and Chao1 indices, were observed for the group of patients at risk compared to patients without clinical symptoms (Figure 3).

Based on CBCL_TOT, the α-diversity, based on Simpson index, showed a higher bacterial diversity (*p*-value ≤ 0.05) in ASD patients with clinical symptoms than in groups at risk for behavioral problems (Figure 3).

On the contrary, based on the CSS, CBCL_INT and IQ/DQ scores, the α-diversity revealed no statistically significant differences between ASD subgroups (Appendix A).

The β-diversity analysis, measured by Bray–Curtis distance, highlighted the absence of significant differences between groups of ASD patients stratified by all neuropsychological parameters (Appendix A).

Examining the GM compositions of ASD patients at different taxonomic levels, specifically at L2, phylum, and L6, genus, dissimilarities were detected between patient subgroups stratified by neuropsychological variables (Appendix A). In particular, at the phylum level, Verrucomicrobiota showed a statistically significantly higher value (*p*-value = 0.035) for the CBCL_INT-based “at risk” subgroup (Appendix A). At the genus level, *Akkermansia* distribution was statistically significant higher (*p*-value = 0.033) for the CBCL_INT-based “symptoms” subgroup (Appendix A).

Both the LEfSe algorithm and linear regression models, the latter corrected by the confounding factor age, were exploited to identify GM bacterial biomarkers associated with ASD, based on all neuropsychological-dependent subgroups. A comparison between clinical CSS categories, namely, mild–moderate and severe autism, revealed *Veillonella_A* genus as the only discriminatory “key” taxon, which was shown to be significantly higher (*p*-value = 0.031) for the severe compared to the mild–moderate ASD group (Appendix A). Considering the CBCL_INT score, no significant differences were detected amongst the three patient subgroups.

On the contrary, by stratifying ASD patients by CBCL EXT, a statistically significant increase in *Clostridium* (sp. CAG-217), *Clostridium*_T and *Eggerthella* (sp. CAG-1427) was observed for the patient group “at risk” for behavior problems compared to the group “without clinical symptoms” (Figure 4). Also, a significantly higher abundance of *Blautia*_A_141781 was observed in the ASD “at risk” group than in the ASD “with clinical symptoms” group; moreover, *Intestinibacter* and *Terrisporobacter* were shown to be more abundant in the group “at risk” compared to both groups “with” and “without clinical symptoms” (Figure 4). The genus *Oscillibacter* (sp. ER4) was associated with both “at risk” and “with clinical symptoms” groups, with higher abundance compared to the “without symptoms” ASD group (Figure 4). Considering the group “with clinical symptoms”, the genera *Bariatricus*, *Coprococcus*_A_121497, *Onthenecus* and *Ruminococcus_D* were over-expressed compared to the group “without clinical symptoms” (Figure 4); a higher abundance of *Faecousia*, belonging to the Oscillospiraceae family, was found in patients “with clinical symptoms” compared to both “at risk for behavior problems” and “without clinical symptoms” groups (Figure 4).

Considering the CBCL_TOT, *Oscillibacter* (sp. ER4), amongst the phylum Firmicutes_A, appeared to increase in the ASD group “with clinical symptoms” compared to both “without clinical symptoms” and “at risk for problems” groups (*p*-value = 0.006) (Appendix A).

No significant statistical comparisons were made based on the IQ/DQ feature.

Considering CSS categories, and focusing on the species taxonomic level, a decrease in *Monoglobus pectinilyticus* was observed in the “severe autism” group (Table 1).

Differences were also found in terms of the distribution of bacteria at the species level when comparing CBCL groups. Specifically, for CBCL_INT, a statistically significant enrichment in the species *Bacteroides_H xylanisolvens* was found in both “with clinical symptoms” and “at risk for behavior problems” ASD groups, compared to the group “without clinical symptoms” (Table 1).

Regarding CBCL_EXT, an over-abundance of *CAG*-217 sp000436335 and *Gemmiger*_A_73129 *qucibialis* in “at risk” patients compared to patients “without clinical symptoms” was seen (Table 1); the group of patients “at risk” also showed a significant increment in the species *Intestinibacter bartlettii* compared to both groups “with” and “without clinical symptoms” (Table 1).

Higher abundances of *Coprococcus*_A_121497 *eutactus*, *Dysosmobacter* sp000403435, *Oscillibacter* ER4 sp900317525, *Oscillibacter ER4* sp000765235 and *Ruminococcus*_D *bicirculans* were detected in patients with “clinical symptoms” compared to patients “without symptoms” based on CBCL_EXT (Table 1).

Finally, considering CBCL_TOT, patients “with clinical symptoms” showed statistically significantly increased distributions of *Oscillibacter ER4 sp900317525* compared to patients “without clinical symptoms” (Table 1). A statistically significant increase in *Anaerotruncus colihominis* was seen in the group of patients “without clinical symptoms” compared to in patients “with clinical symptoms” (Table 1). No statistically significant comparison, based on the IQ/DQ feature, was highlighted at the species level.

### 3.4. Functional Prediction of GM by PICRUSt

The PICRUSt analysis was performed to predict differential KEGG pathways for clinical ASD groups. Only one metabolic pathway was found between the two CSS categories; in particular, the Thiamine metabolism was associated with severe autism (Appendix A).

Considering the CBCL_INT, only the non-homologous end-joining pathway, belonging to the replication and repair pathway class, was associated with the group “at risk” for behavioral problems (Appendix A).

Regarding CBCL_EXT, the group of patients “without clinical symptoms” showed an increase in the metabolism pathway class, especifically for: Carbohydrate metabolism (Glyoxylate and dicarboxylate metabolism); Amino acid metabolism (Tyrosine metabolism); and Xenobiotics biodegradation and metabolism (Metabolism of xenobiotics by cytochrome P450 and Drug metabolism—cytochrome P450). Patients “with clinical symptoms” showed a significant increment in the three pathways belonging to the metabolism pathway class (Terpenoid backbone biosynthesis; Peptidoglycan biosynthesis; and Pyrimidine metabolism) and the three pathways associated with the Genetic Information Processing class (Translation; Aminoacyl-tRNA biosynthesis and ribosome; Transcription RNA polymerase).

Only the pathway of Purine metabolism was associated with “at risk” ASD (Appendix A). For the CBCL_TOT, patients “without clinical symptoms” showed an increase in the Pyrimidine metabolism pathway, while the “at risk” group was associated with the Flavonoid biosynthesis, and patients “with clinical symptoms” presented an increase in Biofilm formation—*Vibrio cholerae* pathway (Appendix A).

Finally, considering the score of IQ/DQ, functional analysis revealed six pathways associated with patients with CI/DD and two associated with patients without CI/DD. The majority of the pathways associated with the group with CI/DD belonged to the metabolism class, in particular with the biosynthesis of other secondary metabolites (Tropane, piperidine and pyridine alkaloid biosynthesis); energy metabolism (Nitrogen metabolism); lipid metabolism (Arachidonic acid metabolism); metabolism of cofactors and vitamins (Vitamin B6 metabolism); and metabolism of other amino acids (Glutathione metabolism). Only one belonged to the endocrine system class, specifically to the PPAR signaling pathway (Figure 5). For the group without CI/DD, one pathway was related to carbohydrate metabolism (Glycolysis/Gluconeogenesis) and the other to signal transduction (MAPK signaling pathway—yeast) (Figure 5).

### 3.5. Correlation between MDI and Neuropsychological Data

Investigating the correlation between MDI and CSS, an increase in MDI in ASD patients with severe autism compared to the mild-moderate autism group was observed (*p*-value > 0.05) (Figure 6). Considering CBCL_INT, the group of patients “with clinical symptoms” showed a higher median value of MDI than both groups of patients “at risk” and “without clinical symptoms” (*p*-value > 0.05) (Appendix A).

On the contrary, regarding CBCL_EXT, patients “without clinical symptoms” showed an increase (*p*-value > 0.05) in MDI compared to patients “at risk” and to patients “with clinical symptoms” (Appendix A).

When considering CBCL_TOT, the group of patients “at risk” displayed the highest value of MDI, followed by the group of patients “with clinical symptoms”. The patients “without clinical symptoms” showed a lower (*p*-value > 0.05) MDI value than those presented by the other two groups (Appendix A).

Considering the IQ/DQ variable, there were no significant differences (*p*-value > 0.05) in MDI value between the two groups.

## 4. Discussion

Given the increasing evidence highlighting the significance of GM in the gut–brain axis and its relationship with neuropsychological ASD phenotypes, identifying essential microbial contributors could impact therapeutic intervention targets in neurological disorders [38].

While the connection between the GM and ASD is intricate and multifaceted, this study aims to focus on data on behavior, anxiety, social responsiveness, challenging behavior, developmental regression parameters, autism severity symptoms, the GM dysbiosis index and related metataxonomy-based profiling.

In our study, ASD children with severe autism symptoms displayed a GM profile characterized by a higher abundance of *Veillonella* when compared to the GM of children with mild to moderate ASD, adding value to previous evidence reporting an increase in *Veillonella* in ASD patients compared to neurotypical subjects in Chinese children [39]. *Veillonella* appears to play a significant role in the development of microbiota and may contribute to microbiota immaturity in young patients with ASD [40].

Interestingly, in our study, consistently with Li et al. [39], this bacterium was detected principally in younger children (age ≤ 6 years). Moreover, in our study, the patients with severe autism showed a decrease in *Monoglobus pectinilyticus* if compared to patients with mild to moderate autism. The *M.*, *pectinilyticus*, a pectin-degrading specialist bacterium, was associable to the commensal GM of healthy subjects [41] and specifically of healthy children [42].

Regardless of the absence of statistical significance, MDI also appeared higher in children with severe forms of the disease, and α- and β-diversity underlined differences among ASD patients, suggesting symptom-specific microbial signatures.

Interestingly, considering CBCL_EXT, an overabundance of the intestinal pathobiont *Clostridium* spp. was detected in the patients at risk for behavior problems, once compared to patients without clinical symptoms. This trend is displayed in other studies that show the potential correlation between Clostridia and the progress of autism symptoms [43].

Moreover, *Intestinibacter* was also more abundant in patients at risk for behavior problems, compared to both subject groups with and without clinical symptoms, while *Blautia* was higher in at-risk patients, compared to patients with clinical symptoms. Several studies detected *Intestinibacter* and *Blautia*, which are closely related to *Clostridium* clusters, as being more abundant in patients with neurodevelopmental disorders (NDD) and ADHD [44,45].

Moreover, a positive association of *Blautia* and *Ruminococcus* with neurological disease severity, such as depression, was reported [46], corroborating the increased value of *Ruminococcus* reported in our study for patients with clinical symptoms. Recently, Monoglobabaceae, Clostridia_UCG-014, Eggethellaceae have also been characterized in children with severe autism [47], consistently with our results.

In our cohort, *Terrisporobacter* was increased in at-risk patients. *Terrisporobacter* has already been reported as higher in ASD children, and has been correlated with tryptophan and pro-inflammatory serum cytokines [48]. Increased levels of these cytokines have been reported in association with a regressive autism phenotype with significant communication challenges and aberrant behaviors [49,50].

In our at-risk patients and those with symptoms, *Oscillibacter* ER4 was increased. *Oscillibacter* ER4 has been already described in ASD patients, appearing positively correlated with fatty acid levels and drugs [51].

*Coprococcus*_A_121497 was more abundant in ASD patients with clinical symptoms. Different evidence has linked *Coprococcu*s to multiple neurological and psychological disorders such as depression, Parkinson’s disease, epilepsy, obsessive and compulsive disorders and schizophrenia, establishing a link between these taxa and the gut–brain axis [52].

Regarding *Bariatricus*, *B. massiliensis* was recently identified as a new bacterial species from human GM in a patient operated by bariatric surgery [53].

The correlation analysis between GM ecology and KEGG pathways provided valuable insights into potential links between bacterial taxa and functional ASD metabolic pathways. Particularly, for the CBCL_INT group at risk for problematic behavior, the non-homologous end-joining pathways were expressed as unique pathways for this group. This pathway is involved in a pivotal double-strand break repair mechanism and closely associated with various conditions, including aging, neurodegenerative diseases, inflammation, obesity, diabetes mellitus, and cancer [54,55]. Recent studies have demonstrated that probiotic supplementation can inhibit the Non-homologous end-joining pathway leading to a reduction in inflammatory responses [56], as reported also for schizophrenic patients [57].

The CBCL_EXT patients without clinical symptoms showed specific over-expressed pathways, particularly the glyoxylate and dicarboxylate metabolism, which is essential for organism survival and biochemical functions. In particular, within the glyoxylate pathway, citrate synthase (K01647) catalyzes the formation of citrate from acetyl-CoA and oxaloacetate. Subsequently, citrate is transformed into succinate, which plays a crucial role in glucose synthesis [58,59]. This process involves the conversion and use of intermediates, produced from the catabolism of fatty acids, the metabolism of amino acids, and the fermentation of carbohydrates [60].

Moreover, these CBCL_EXT patients without clinical symptoms were characterized by the over-expression of the aromatic amino acids (AAAs) pathways. The metabolites derived from AAAs play a role in regulating the host’s local and systemic immune, metabolic, and neuronal responses. Moreover, these metabolic pathways are linked to a range of diseases, including those affecting the central nervous system [61].

On the contrary, the CBCL_EXT patient group with clinical symptoms was characterized by an over-expression of terpenoid backbone biosynthesis that generates steroids, which are subsequently transformed into cholesterol through the steroid biosynthesis route. Cholesterol is then converted into cholesteryl esters via esterification with long-chain fatty acids [62]. These cholesteryl esters interact with triacylglycerol to form very-low-density lipoproteins (VLDLs) that are significantly associated with coronary artery calcification [63]. Also, the pyrimidine metabolism was over-expressed. This pathway influences the regulation of colonic inflammation in mice, encompassing the synthesis, interconversion, salvage, and pyrimidine breakdown nucleotides in humans [64].

The patients at risk for CBCL_TOT were associated with flavonoid biosynthesis. This metabolic pathway generates flavonoids, and related aromatic bacterial metabolites have a considerable impact on the intestinal environment, potentially influencing the physiology and biochemistry of the gut, by modulating inflammatory-related cellular events in the intestine and/or the composition of the microbiota populations, particularly during metabolic disturbances [65].

On the contrary, the group with CI/DD was characterized by lipid metabolism, which is generally associated with microbiota dysbiosis [66]. In particular, chronic stress disrupts the lipid metabolism and leads to diminished neurogenesis. Moreover, the GM may impact the host’s inflammatory metabolic and energy metabolism pathways by affecting glucose and lipid metabolism [67].

Regarding the amino acid metabolism abnormalities detected in CI/DD, significant changes in various tyrosine and phenylalanine derivatives [68] related to dopamine synthesis and metabolism have been shown. Dopamine has been identified as one of the main neurotransmitters for social behavior, social cognition, and the control of movement [69]. Generally, certain GM bacteria exhibit a strong correlation with these metabolites, suggesting that bacteria, such as *Coprococcus*_A_121497, may contribute to dopamine signaling abnormalities by influencing the amino acid metabolism [51].

Lastly, the over-expression of the glutathione metabolism in the group with CI/DD may be crucial in the pathogenesis of ASD [70]. Indeed, Frustaci et al. [71] have reported systemic deficiencies in glutathione and cysteine in individuals with ASD.

## 5. Conclusions

The investigation of the GM profile in relation to ASD neuropsychological symptoms and MDI may represent a promising tool to define gut microbial dysbiosis and new microbial signatures associated with ASD phenotypes, especially severity. However, the limitation of the small patient cohorts needs to be overcome to ensure GM profiling strength and validation across growing numbers of different sample datasets. Once broader datasets are made available, GM ecology, MDIs and functional correlations with neuropsychological features can be incorporated into a new strategy that will contribute to an integrated diagnostic assessment of ASD.

## Figures and Tables

**Figure 1 microorganisms-12-02041-f001:**
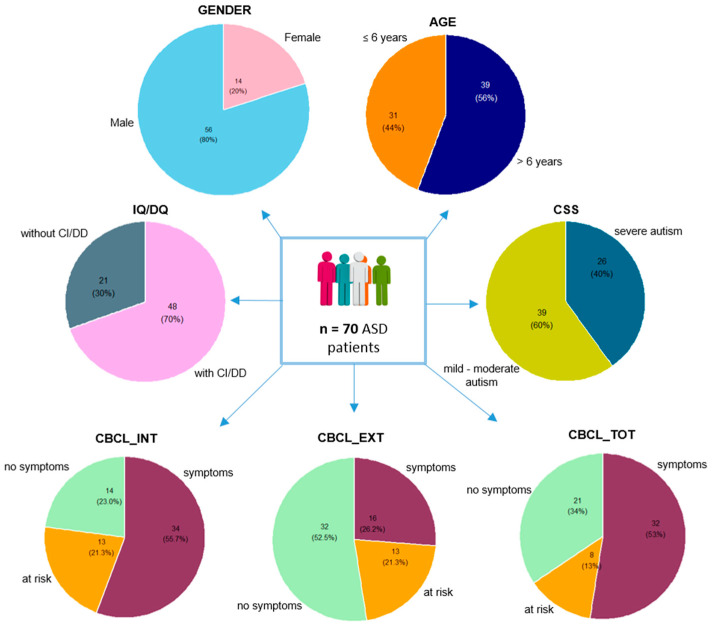
Stratification representation of ASD patients based on anamnestic and neuropsychological features.

**Figure 2 microorganisms-12-02041-f002:**
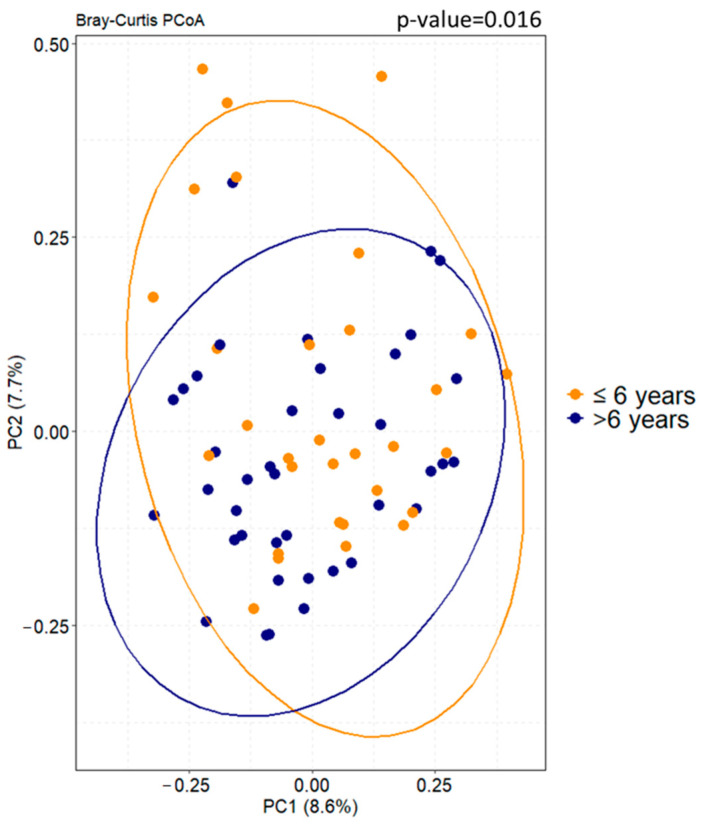
Beta diversity analysis based on Bray–Curtis dissimilarity algorithm calculated for ASD patients stratified by the confounding factor of age. PERMANOVA *p*-value is reported.

**Figure 3 microorganisms-12-02041-f003:**
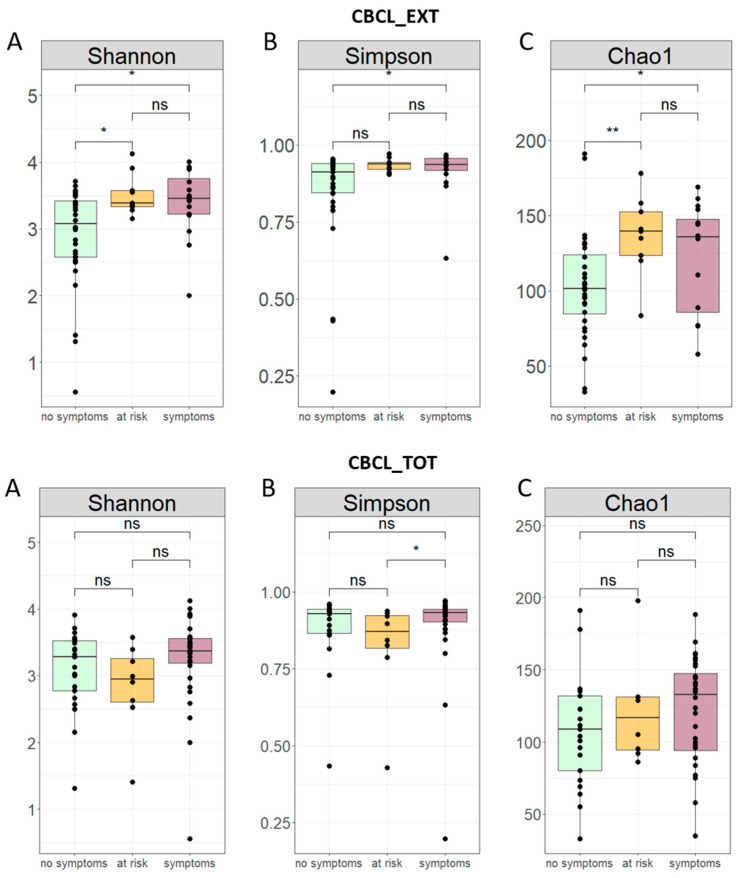
Alpha diversity of ASD patients grouped by CBCL_EXT and CBCL_TOT. Alpha diversity representation of ASD patients grouped by CBCL_EXT and CBCL_TOT based on Shannon (**A**), Simpson (**B**) and Chao1 (**C**) indices. Comparisons amongst the three subgroups (no symptoms; at risk; and symptoms) were performed by Kruskal–Wallis tests, while the sub-group pairwise comparisons were performed through a post-hoc Mann–Whitney test (ns: *p*-value > 0.05; ** *p*-value ≤ 0.01; * *p*-value < 0.05).

**Figure 4 microorganisms-12-02041-f004:**
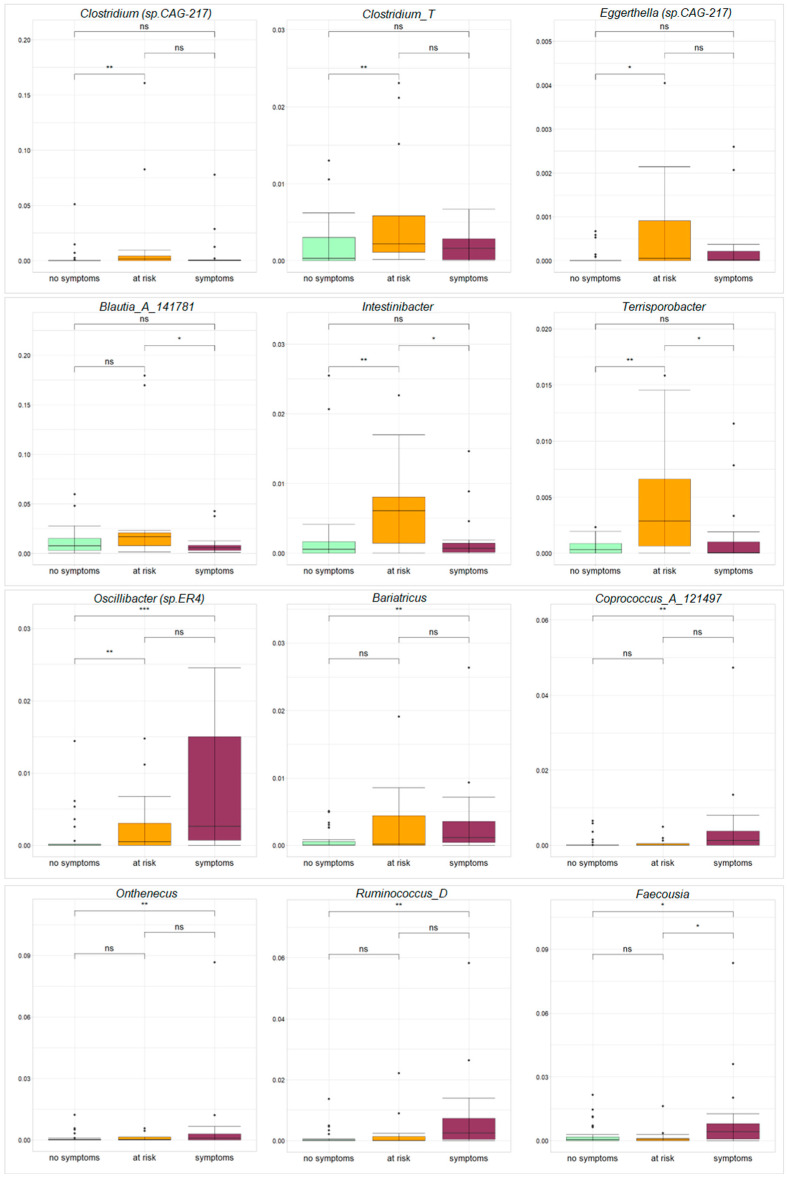
Microbial biomarkers discriminating ASD patients grouped by CBCL_EXT. Distribution of statistically significant genera for ASD patients stratified by CBCL_EXT (Kruskal–Wallis test, *p*-value < 0.05). The pairwise comparisons were obtained by a post hoc Mann–Whitney test (ns: *p*-value > 0.05; *** *p*-value ≤ 0.001; ** *p*-value ≤ 0.01; * *p*-value < 0.05).

**Figure 5 microorganisms-12-02041-f005:**
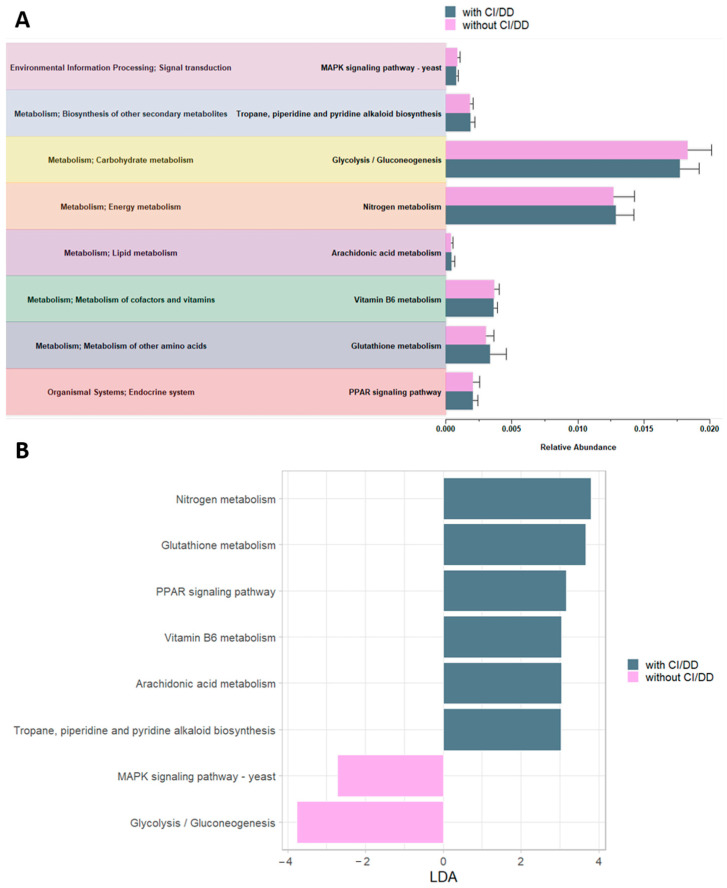
Prediction of functional KEGG pathways performed by PICRUSt2 to discriminate ASD patients grouped by IQ/DQ. Statistically significant functional pathways were identified for ASDs patients stratified based on the IQ/DQ feature by the Limma Voom method (**A**) and LEfSe with LDA > 2 (**B**) (*p*-value < 0.05).

**Figure 6 microorganisms-12-02041-f006:**
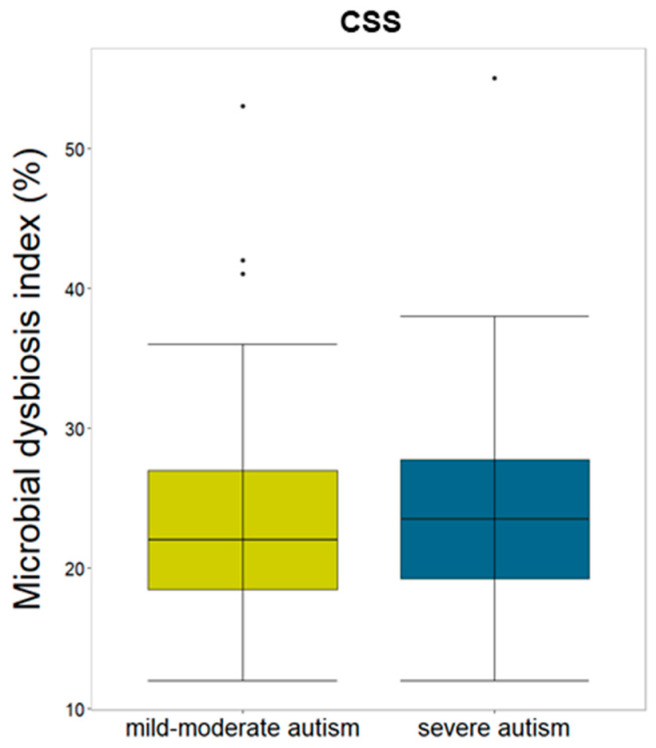
MDI distribution for ASD patients grouped by CSS. Boxplot of MDI distribution (percentage) in patients stratified by CSS.

**Table 1 microorganisms-12-02041-t001:** Microbial species biomarkers of ASD clinical subgroups based on neuropsychological features.

CSS							
Phylum	Family	Genus	Species	Mild–Moderate Autism (Mean Value)	Severe Autism (Mean Value)		Mann–Whitney Test (*p*-Value)
Firmicutes_A	Monoglobaceae	*Monoglobus*	*pectinilyticus*	0.0028	0.0005		0.04
**CBCL_INT**							
**Phylum**	**Family**	**Genus**	**Species**	**no symptoms** (mean value)	**at risk** (mean value)	**symptoms** (mean value)	**Kruskal test** (*p*-value)
Bacteroidota	Bacteroidaceae	*Bacteroides_H*	*xylanisolvens*	0.0005	0.0088	0.0034	0.0200
**CBCL_EXT**							
**Phylum**	**Family**	**Genus**	**Species**	**no symptoms** (mean value)	**at risk** (mean value)	**symptoms** (mean value)	**Kruskal test** (*p*-value)
Firmicutes_A	Acutalibacteraceae	*CAG-217*	*sp000436335*	0.0024	0.0205	0.0076	0.0200
	Lachnospiraceae	*Coprococcus_A_121497*	*eutactus*	0.0006	0.0006	0.0051	0.0400
	Oscillospiraceae_88309	*Dysosmobacter*	*sp000403435*	0.0004	0.0013	0.0031	0.0200
	Oscillospiraceae_88309	*ER4*	*sp000765235*	0.0009	0.0009	0.0047	0.0020
	Oscillospiraceae	*ER4*	*sp900317525*	0.0002	0.0021	0.0034	0.0002
	Ruminococcaceae	*Gemmiger_A_73129*	*qucibialis*	0.0118	0.0469	0.0166	0.0050
	Peptostreptococcaceae_256921	*Intestinibacter*	*bartlettii*	0.0023	0.0064	0.0022	0.0060
	Ruminococcaceae	*Ruminococcus_D*	*bicirculans*	0.0013	0.0028	0.0082	0.0080
**CBCL_TOT**							
**Phylum**	**Family**	**Genus**	**Species**	**no symptoms** (mean value)	**at risk** (mean value)	**Symptoms** (mean value)	**Kruskal test** (*p*-value)
Firmicutes_A	Oscillospiraceae	*ER4*	*sp900317525*	0.0003	0.0003	0.0025	0.0070
	Ruminococcaceae	*Anaerotruncus*	*colihominis*	0.0036	0.0017	0.0003	0.0200

## Data Availability

The datasets presented in this study can be found in online repositories. The names of the repository and accession number can be found below: PRJNA1136218 (https://www.ncbi.nlm.nih.gov/bioproject, accessed on 6 October 2024).

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
