# Peer review of "Stratification of Gut Microbiota Profiling Based on Autism Neuropsychological Assessments"

_microorganisms, 2024, doi:10.3390/microorganisms12102041_

Round 1

Reviewer 1 Report

Comments and Suggestions for Authors

The manuscript by Marangelo C. et al., reports an interesting study on the correlation between gut microbiota and autism disease in children and adolescents.

The introduction is well written and give to the readears a wide picture of the topic.

Methods section described in detail all methodologies used in the present study, statistical methods are appropriate.

Results are detailed and very clear. Discussion well correlate the results obtained with literature.

Author Response

Comments and Suggestions for Authors Review 1

The manuscript by Marangelo C. et al., reports an interesting study on the correlation between gut microbiota and autism disease in children and adolescents.

The introduction is well written and give to the readears a wide picture of the topic.

Methods section described in detail all methodologies used in the present study, statistical methods are appropriate.

Results are detailed and very clear. Discussion well correlate the results obtained with literature.

AUTHORS’ REPLY.

We thank the Reviewer 1 for her/his kind and careful examination of our manuscript and we gratefully acknowledge for the positive evaluation of our work and the comments.

Reviewer 2 Report

Comments and Suggestions for Authors

The study entitled “Stratification of gut microbiota profiling based on autism neuropsychological assessments” includes 70 autism spectrum disorder patients was conducted to assess gut microbiota to understand the severity and symptoms of the neurodevelopmental disorder. Authors observed that 60% of autism spectrum disorder patients had mild-to-moderate autism, while 40% had severe symptoms. As per the authors the gut microbiota of severe autism spectrum disorder patients in the study was characterized by increased Gram-negative bacteria anaerobic cocci, Veillonella, and decreased Monoglobus pectinilyticus, and a higher microbial dysbiosis index compared to mild-moderate autism spectrum disorders. Patients at risk for behavior problems and with clinical symptoms had an increased gut microbiota of Clostridium, Eggerthella, Blautia, Intestinibacter, Coprococcus, Ruminococcus, Onthenecus, and Bariatricus. Peptidoglycan biosynthesis and Biofilm formation related pathways were identified in patients with clinical symptoms, and potential microbiota-activated PPAR-γ-signaling cognitive impairment/developmental delay patients were also identified. Authors feel that the observed findings could improve understanding of autism spectrum disorder and enhance neurological phenotype comprehension.

One of the primary concerns is related to the gender ratio. While we can understand that certain conditions may be more prevalent in one sex than the other its essential for researchers to maintain a balanced representation of both genders in their studies. There is a significant imbalance (eg more than 75% of participants are from one gender), researchers should provide a clear justification or acknowledgment of this potential bias. This ensures that the findings are applicable to a broader population and minimizes the risk of gender-specific biases influencing the results.

The majority of the findings look interesting. However, the other should concentrate on precise writing while they are writing the content in the results most of the time they didn't mention the P values, whatever they got in their result. For example “Finally, considering CBCL_TOT, patients “with clinical symptoms” had statistically significantly increased distributions of Oscillibacter ER4 sp900317525, compared to patients “without clinical symptoms”. A statistically significant increase of Anaerotruncus colihominis was found in the group of patients “without clinical symptoms” than in patients”

Another major concern is related to the sample size of patients. In some groups, the number of participants is insufficient with single digit counts. From a statistical perspective, such small sample sizes can limit the reliability of the findings. Adequate sample sizes are crucial for drawing meaningful conclusions and ensuring the validity of the gut microbiota profiling.

Comments on the Quality of English Language

Moderate editing of English language required

Author Response

Comments and Suggestions for Authors Reviewer 2

The study entitled “Stratification of gut microbiota profiling based on autism neuropsychological assessments” includes 70 autism spectrum disorder patients was conducted to assess gut microbiota to understand the severity and symptoms of the neurodevelopmental disorder. Authors observed that 60% of autism spectrum disorder patients had mild-to-moderate autism, while 40% had severe symptoms. As per the authors the gut microbiota of severe autism spectrum disorder patients in the study was characterized by increased Gram-negative bacteria anaerobic cocci, Veillonella, and decreased Monoglobus pectinilyticus, and a higher microbial dysbiosis index compared to mild-moderate autism spectrum disorders. Patients at risk for behavior problems and with clinical symptoms had an increased gut microbiota of Clostridium, Eggerthella, Blautia, Intestinibacter, Coprococcus, Ruminococcus, Onthenecus, and Bariatricus. Peptidoglycan biosynthesis and Biofilm formation related pathways were identified in patients with clinical symptoms, and potential microbiota-activated PPAR-γ-signaling cognitive impairment/developmental delay patients were also identified. Authors feel that the observed findings could improve understanding of autism spectrum disorder and enhance neurological phenotype comprehension.

AUTHORS’ REPLY.

We thank the Reviewer 2 for the comprehensive and thoughtful evaluation of our manuscript. We truly value the positive feedback on our work as well as the insightful suggestions offered. We revised the article accordingly your following suggestions by using the “track change” option in the .docx version.

One of the primary concerns is related to the gender ratio. While we can understand that certain conditions may be more prevalent in one sex than the other its essential for researchers to maintain a balanced representation of both genders in their studies. There is a significant imbalance (eg more than 75% of participants are from one gender), researchers should provide a clear justification or acknowledgment of this potential bias. This ensures that the findings are applicable to a broader population and minimizes the risk of gender-specific biases influencing the results.

AUTHORS’ REPLY.

We thank the Reviewer 2 for highlighting this important issue, and we agree with this observation. This potential bias can be attributed to the higher prevalence of ASD in the male population, as documented by various epidemiological studies, which indicate that males with ASD are four to five times more frequent than females (Salari, N., Rasoulpoor, S., Rasoulpoor, S. et al. The global prevalence of autism spectrum disorder: a comprehensive systematic review and meta-analysis. Ital J Pediatr 48, 112 (2022). https://doi.org/10.1186/s13052-022-01310-w; Developmental Disabilities Monitoring Network Surveillance Year 2010 Principal Investigators; Centers for Disease Control and Prevention (CDC). Prevalence of autism spectrum disorder among children aged 8 years - autism and developmental disabilities monitoring network, 11 sites, United States, 2010. MMWR Surveill Summ. 2014 Mar 28;63(2):1-21. PMID: 24670961). Hence, this imbalance in our work, actually reflects the gender proportion in the real ASD population.

However, to evaluate the weight of the variable "gender", we treated this variable as a potential confounding factor. Specifically, we conducted ecological analysis, including α- and β-diversity assessments, to examine the relationship between GM composition and gender as a covariate. The results indicated no statistically significant findings (Supplementary Figure 1 B-C).

In addition, we performed, at L2, L6 and L7 taxonomic levels, a univariate analysis (Mann-Whitney test) to identify potential biomarker associated to the variable “gender”. Only the “Mediterraneibacter_A_155507” genus and the “Mediterraneibacter_A_155507_faecis” species resulted statistically significant different (p-value=0.019 and p-value=0.027, respectively). In particular these genus and species showed higher values in female group compared to male group. However, as these taxa were not found to be biomarkers in all other analyses, we can infer that gender is not a confounding factor for our results.

In order to clarify also to the readers, we have reported the new following sentence in the “3.2. Metataxonomic GM profile” section, text R1 version: “The ASD cohort was categorized by age and gender to assess their potential influence as confounding factors on the GM profile” and “Conversely, considering the gender variable, both ecological and univariate analyses revealed no statistically significant results, indicating that gender was not a confounding factor (Supplementary Figure 1).

Supplementary Figure 1. Ecological analyses of ASD patients grouped by potential confounding factor age and gender. Alpha diversity based on Shannon, Simpson and Chao1 indices of ASD patients grouped by age (A) and gender (B). The pairwise-comparisons were obtained by a post hoc Mann–Whitney test (ns: p-value > 0.05).

Beta-diversity analysis based on Bray–Curtis dissimilarity algorithm calculated for ASD patients stratified by gender (C). PERMANOVA p-value is reported.

The majority of the findings look interesting. However, the other should concentrate on precise writing while they are writing the content in the results most of the time they didn't mention the P values, whatever they got in their result. For example “Finally, considering CBCL_TOT, patients “with clinical symptoms” had statistically significantly increased distributions of Oscillibacter ER4 sp900317525, compared to patients “without clinical symptoms”. A statistically significant increase of Anaerotruncus colihominis was found in the group of patients “without clinical symptoms” than in patients”

AUTHORS’ REPLY.

We thank the Reviewer 2 for this comment. Following your suggestion, we added the p values in the text and we clarified the reference to Figures/Tables containing p-values in the “Results” section, as requested.

Another major concern is related to the sample size of patients. In some groups, the number of participants is insufficient with single digit counts. From a statistical perspective, such small sample sizes can limit the reliability of the findings. Adequate sample sizes are crucial for drawing meaningful conclusions and ensuring the validity of the gut microbiota profiling.

AUTHORS’ REPLY.

We thank the Reviewer 2 for this comment. We are aware that the limited sample size of specific patient subgroups presents a constraint. However, we believe that our cohort is composed by a number of patients (n= 70) limited but still interesting to explore various clinical severity indices, especially if compared with previous studies in the field.

This manuscript represents the first effort to correlate ASD-related neurological features with GM composition, aiming to identify potential biomarkers.

In the next future, the goal will be to expand the sample size to enhance statistical robustness, as noted in the conclusions: " However, the limitation of small patient cohorts needs to be overcome, to ensure GM pro-filing strength and validation, across growing numbers of different sample datasets."

Round 2

Reviewer 2 Report

Comments and Suggestions for Authors

The authors have tried to incorporate the suggestion in the first round of review.

The revised version can be considered for publication.

Comments on the Quality of English Language

Minor editing of English language required.